# Evaluation of Direct Antimicrobial Susceptibility Testing of Gram-Negative Bacilli and *Staphylococcus aureus* from Positive Pediatric Blood Culture Bottles Using BD Phoenix M50

**DOI:** 10.3390/microorganisms12081704

**Published:** 2024-08-18

**Authors:** Princess Morales, Patrick Tang, Elaine Mariano, Arun Gopalan, Nisha Aji, Andrés Pérez-López, Mohammed Suleiman

**Affiliations:** 1Department of Pathology, Sidra Medicine, Doha P.O. Box 26999, Qatar; pmorales@sidra.org (P.M.); ptang@sidra.org (P.T.); emariano@sidra.org (E.M.); agopalan@sidra.org (A.G.); naji@sidra.org (N.A.); aperezlopez@sidra.org (A.P.-L.); 2Department of Pathology and Laboratory Medicine, Weill Cornell Medicine in Qatar, Doha, P.O Box 24144, Qatar

**Keywords:** blood stream infections, direct antimicrobial susceptibility testing, pediatric blood cultures, *Staphylococcus aureus*

## Abstract

Bloodstream infections (BSIs) are life-threatening infections for which a timely initiation of appropriate antimicrobial therapy is critical. Antibiotic susceptibility testing (AST) directly performed on positive blood culture broths can help initiate targeted antibiotic therapy sooner than the standard AST performed on colonies isolated on solid media after overnight incubation. Faster antimicrobial susceptibility testing (AST) results can improve clinical outcomes, and reduce broad-spectrum antimicrobial consumption and healthcare-associated costs in sepsis. In this study, we evaluated the accuracy of a direct AST inoculation method on the BD Phoenix M50 system using serum separator tubes to harvest bacteria from positive pediatric blood culture bottles. Direct AST was performed on 132 monomicrobial pediatric blood culture bottles that were positive for Enterobacterales (65; 49.2%), *Staphylococcus aureus* (46; 34.8%), and non-fermenting Gram-negative bacilli (21; 16%). Overall, the categorical and essential agreements between the direct method and standard method were 99.6% and 99.8%, respectively. Very major, major, and minor error rates were 0.1%, 0.09%, and 0.20% respectively. Direct AST performed on pediatric blood culture bottles using BD Phoenix M50 can quickly provide accurate susceptibility information to guide antimicrobial therapy in patients with BSI.

## 1. Introduction

Faster identification and antimicrobial susceptibility testing (AST) results can improve clinical outcomes, and reduce broad-spectrum antimicrobial consumption and healthcare-associated costs in sepsis [1,2,3,4]. Traditional AST for bacteria recovered from positive blood cultures requires an overnight subculture of an aliquot inoculated on agar plates, which typically takes 24 to 72 h to yield results. In contrast, direct AST can be carried out using a serum separator tube (SST) for harvesting the bacteria directly from positive blood culture broths, allowing susceptibility results within the first 24 h after bottles are removed from the automated system [5,6,7,8,9,10]. This method has shown reliable performance in previous studies, demonstrating a positive impact on patient care by providing physicians with faster AST results at least one day earlier than traditional methods [5,6,7,8,9]. Although this approach has been evaluated using several automated systems in adult blood culture bottles, it has never been assessed using the BD Phoenix M50 system in the pediatric population. Here, we prospectively evaluated the accuracy of a direct AST method performed on positive pediatric blood cultures collected from children with sepsis at Sidra Medicine.

## 2. Materials and Methods

### 2.1. Study Design, Sample Collection, and Blood Culture Specimen Processing

Between May 2023 and July 2024, this prospective study was conducted in the Department of Microbiology at Sidra Medicine, a pediatric tertiary care hospital in the State of Qatar. Blood drawn from pediatric patients suspected of BSI was inoculated in blood culture bottles (BD BACTEC™ Peds Plus/F) (Becton, Dickinson, and Company, Franklin Lakes, NJ, USA) and monitored for bacterial growth in the BD BACTEC FX blood culture system (Becton, Dickinson, and Company, Franklin Lakes, NJ, USA). When the blood culture bottles were flagged as positive by the instrument, Gram staining was performed, and aliquots were plated onto the appropriate agar media. Additionally, direct organism identification was performed using our in-house developed method for matrix-assisted laser desorption ionization–time of flight mass spectrometry (MALDI-TOF MS) (Bruker, Bremen, Germany) [11]. Acceptable samples in this study included positive pediatric blood cultures with only one organism seen on Gram stain, and a direct identification of either *Staphylococcus aureus* or a Gram-negative organism. Positive blood cultures from adult patients, those positive with other organisms, or polymicrobial positive pediatric blood cultures were excluded from this study.

### 2.2. AST Using the Standard Method

A small volume of the positive blood culture bottle was inoculated on a sheep blood agar plate and incubated for 18–24 h at 35 °C with 5% CO_2_. A standard inoculum was prepared in the Phoenix ID broth (Becton, Dickinson, and Company, Franklin Lakes, NJ, USA) from pure isolated colonies grown on the blood agar plates and inoculated into Phoenix AST panels (Becton, Dickinson, and Company, Franklin Lakes, NJ, USA), following the manufacturer’s instructions. For the Gram-negative organisms, the NMIC-501 Phoenix panel was used, and the PMIC-111 Phoenix panel was used for *S. aureus*.

### 2.3. AST Using Direct Inoculation

From the positive blood culture bottle, 3.5 mL was inoculated into a BD serum separator tube (SST) (Becton, Dickinson, and Company, Franklin Lakes, NJ, USA). The SST was then centrifuged for 10 min at 2000× *g* and the supernatant was discarded. The bacteria were harvested from the gel layer using a sterile cotton swab and resuspended into the Phoenix ID broth tube until a 0.5 McFarland standard suspension was obtained. The prepared bacterial inoculum was then transferred to appropriate Phoenix panels following the manufacturer’s instructions. A visual comparison showing the difference between the standard and direct AST methods is shown in Figure 1.

### 2.4. Conflict Resolution and Statistical Analysis

A minimum inhibitory concentration (MIC) test strip (Liofilchem, Roseto degli Abruzzi, Italy) was performed according to the manufacturer’s instructions to resolve discrepancies between direct inoculation and the standard method. AST results obtained by the standard method and direct inoculation method were analyzed for categorical agreement (CA) and essential agreement (EA). CA was defined as complete agreement if both the standard and direct method susceptibility categories (susceptible, intermediate, resistant) were the same. EA was defined as complete agreement if the MICs of the standard and direct methods were within ±1 dilution. Any disagreement between the direct AST and the standard AST method was further categorized as minor error (mE) for false intermediate results, major error (ME) for false resistant results, and very major error (VME) for false susceptible results. MICs and breakpoints were interpreted according to the 2023 Clinical and Laboratory Standards Institute (CLSI) guidelines [12].

## 3. Results

### 3.1. Samples Tested

A total of 132 positive blood cultures were analyzed, corresponding to 3294 antimicrobial combinations. Sixty-five (49.2%) isolates were Enterobacterales, 46 (34.8%) *Staphylococcus aureus*, and 21 (16%) non-fermenting Gram-negative bacteria (Table 1).

### 3.2. AST Results

The standard AST and direct AST methods showed an overall CA of 99.6% and EA of 99.8%. The analysis resulted in 14 errors, 3 (0.1%) VMEs, 3 (0.09%) MEs, and 8 (0.20%) mEs (Table 1 and Table 2). Thirteen errors (92.9%) were recorded among Enterobacterales, particularly *Klebsiella pneumoniae* (8; 62%) (Table 1). The essential agreement rates for all isolates were 100%, except for *K. pneumoniae* (99.3%) and *S. aureus* (99.9%) (Table 1). As for non-fermenting Gram-negative bacteria, the EA and CA were 100% and no errors were observed (Table 1). The EA was 100% for all antibiotics except ertapenem, fosfomycin, nitrofurantoin, tigecycline against *K. pneumoniae*, and trimethoprim–sulfamethoxazole (SXT) against *S. aureus* (Table 2). Two VMEs for nitrofurantoin and one for SXT were observed (Table 2). Eight mE were observed among Enterobacterales for ampicillin–sulbactam, cefazolin, ceftriaxone, cefuroxime, ciprofloxacin, nitrofurantoin, and tigecycline (Table 2).

## 4. Discussion

The microbiology laboratory can play a critical role in reducing the time to effective therapy among patients with BSI by performing rapid phenotypic and genotypic assays directly on positive blood culture bottles. To our knowledge, this is the first study evaluating direct AST from positive pediatric blood bottles using the BD Phoenix M50 system. Our findings showed excellent essential and categorical agreement (>99%) compared with the standard AST performed on colonies. In addition, the performance of our method was superior to that observed in studies where direct AST was performed on positive blood cultures in the adult population and had varying categorical agreement (89–99%) when comparing the standard automated method and direct AST method [2,4,5,6,7,8,9,10,13,14,15].

In our study, *S. aureus* was the only Gram-positive species assessed as a BSI caused by this microorganism, and it is associated with high mortality and complication rates in the absence of effective therapy [16]. Noteworthily, the EA and CA rates were 99.9% for *S. aureus*, with only one VME observed for SXT. By contrast, one study also found four VME with SXT by direct inoculation using the Phoenix system in coagulase-negative staphylococci [6]. In addition, categorical agreement for both oxacillin and vancomycin were 100%, with no errors. This finding concurs with previous studies where the direct AST method using the BD Phoenix system [2,6] was assessed in the adult population, suggesting that the direct AST method provides reliable results for the antibiotics considered first-line in the treatment of staphylococcal systemic infections.

Likewise, the direct AST for Gram-negative rods also showed outstanding performance, particularly for non-fermenters, in which all antibiotics tested had 100% essential and categorical agreement. Two very major errors occurred with nitrofurantoin in *K. pneumoniae*. These errors might not have clinical consequences, as nitrofurantoin is only indicated to treat uncomplicated lower urinary tract infection [17].

The direct AST method has several advantages and benefits over standard AST methods. It can be performed as soon as the bottle is flagged positive, and the procedure can be completed in less than 30 min. This is much faster than the traditional method, which involves subculture of the positive blood culture onto agar plates and incubation, which can take up to 24 h. Also, the procedure is user-friendly, with a few steps that are easy to follow, inexpensive, as only an SST is required to be purchased, and can easily be integrated into the laboratory workflow. Conversely, the direct AST method is not recommended in positive blood cultures with multiple pathogens as it can lead to misleading results. It has been reported that 6 to 10% of blood culture specimens that showed one organism on Gram-stain subsequently grew more than two species after subculture to agar plates [4,9]. Since the detection of polymicrobial blood cultures by visualizing a Gram-stain can be challenging, it has been suggested that direct AST results be labeled as “preliminary” until they are confirmed by the standard AST method [14].

A limitation of our study is the low number of samples tested, as we only included select clinically significant species (*S. aureus* and Enterobacterales). Future studies with a larger number of samples and a wider variety of pathogens will be necessary to better evaluate the performance of our method in the pediatric population with suspected BSI. In addition, the clinical impact of this method was beyond the scope of our study, but future studies should explore the clinical benefits of direct AST in the context of an antimicrobial stewardship program.

## 5. Conclusions

In conclusion, our study has demonstrated that the direct AST method using BD Phoenix M50 panels and serum separator tubes to harvest bacteria from positive pediatric blood culture bottles is an accurate and faster diagnostic tool that can help clinicians reduce the time to optimal antimicrobial therapy in patients with BSI. In clinical practice, the adoption of this method has the potential to improve clinical outcomes, reduce broad-spectrum antimicrobial consumption, and lower healthcare-associated costs in sepsis for the pediatric patient population.

## Figures and Tables

**Figure 1 microorganisms-12-01704-f001:**
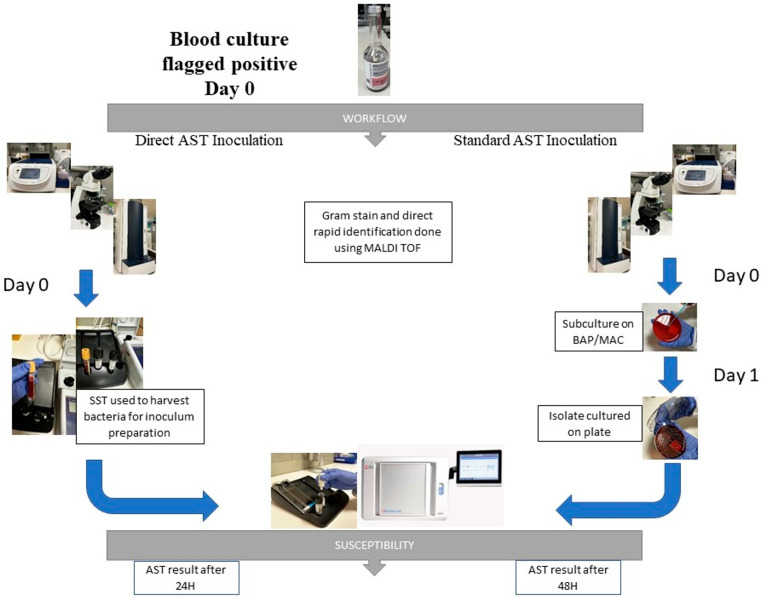
Workflow comparison between standard AST and direct AST methods.

**Table 1 microorganisms-12-01704-t001:** Agreement and error rates in direct AST method per bacterial isolates.

Bacterial Isolates (*n*)	Total Tested	%EA	%CA	mE (%)	ME (%)	VME (%)
Gram-negative Enterobacterales						
*Klebsiella pneumoniae* (28)	728	99.3	98.9	3 (0.4%)	3 (0.4%)	2 (0.3%)
*Klebsiella oxytoca* (1)	26	100	100	0	0	0
*Klebsiella aerogenes* (1)	26	100	100	0	0	0
*Escherichia coli* (15)	390	100	99.4	2 (0.5%)	0	0
*Enterobacter* spp. (5)	130	100	99.2	1 (0.8%)	0	0
*Salmonella* spp. (9)	234	100	99.1	2 (0.8%)	0	0
*Citrobacter* spp. (1)	26	100	100	0	0	0
*Serratia marcescens* (5)	130	100	100	0	0	0
Gram-negative non-fermenters						
*Pseudomonas aeruginosa* (13)	338	100	100	0	0	0
*Acinetobacter* spp. (8)	208	100	100	0	0	0
Gram-positive						
*Staphylococcus aureus* (46)	1058	99.9	99.9	0	0	1 (0.09%)
Total (132)	3294	99.8	99.6	8 (0.20%)	3 (0.09%)	3 (0.09%)

CA, categorical agreement; EA, essential agreement; mE, minor error; ME, major error; n, number of isolates; VME, very major error.

**Table 2 microorganisms-12-01704-t002:** Agreement and error rates in direct AST method per antibiotics.

Antimicrobial Agents	(*n*)	%EA	%CA	mE (%)	ME (%)	VME (%)
Amikacin	86	100	100	0	0	0
Ampicillin	132	100	100	0	0	0
Ampicillin–Sulbactam	86	100	98.8	1 (1.2%)	0	0
Azteronam	86	100	100	0	0	0
Ceftaroline	46	100	100	0	0	0
Cefotaxime	46	100	100	0	0	0
Cefazolin	86	100	98.8	1 (1.2%)	0	0
Cefepime	86	100	100	0	0	0
Cefoxitin	132	100	100	0	0	0
Ceftazidime	86	100	100	0	0	0
Ceftazidime–Avibactam	86	100	100	0	0	0
Ceftolozane–Tazobactam	86	100	100	0	0	0
Ceftriaxone	86	100	98.8	1 (1.2%)	0	0
Cefuroxime	86	100	97.7	2 (2.3%)	0	0
Ciprofloxacin	132	100	99.2	1 (1.2%)	0	0
Clindamycin	46	100	100	0	0	0
Colistin	86	100	100	0	0	0
Daptomycin	46	100	100	0	0	0
Erythromycin	46	100	100	0	0	0
Ertapenem	86	98.8	98.8	0	1 (1.2%)	0
Fosfomycin	86	98.8	98.8	0	1 (1.2%)	0
Gentamicin	132	100	100	0	0	0
Gentamycin-synergy	46	100	100	0	0	0
Imipenem	86	100	100	0	0	0
Levofloxacin	132	100	100	0	0	0
Linezolid	46	100	100	0	0	0
Moxifloxacin	46	100	100	0	0	0
Mupirocin HL	46	100	100	0	0	0
Meropenem	86	100	100	0	0	0
Minocycline	86	100	100	0	0	0
Nitrofurantoin	132	98.4	97.7	1 (0.75%)	0	2 (1.5%)
Norfloxacin	86	100	100	0	0	0
Oxacillin	46	100	100	0	0	0
Penicillin	46	100	100	0	0	0
Piperacillin–Tazobactam	86	100	100	0	0	0
Rifampin	46	100	100	0	0	0
Tetracycline	46	100	100	0	0	0
Teicoplanin	46	100	100	0	0	0
Tigecycline	132	99.2	98.4	1 (0.75%)	1 (0.75%)	0
SXT	132	99.2	99.2	0	0	1 (0.9%)
Vancomycin	46	100	100	0	0	0
Total	3294	99.8	99.6	8 (0.20%)	3 (0.09%)	3 (0.09%)

CA, categorical agreement; EA, essential agreement; HL, high level; mE, minor error; ME, major error; *n,* number of isolates; SXT, trimethoprim–sulfamethoxazole; VME, very major error.

## Data Availability

The original contributions presented in the study are included in the article; further inquiries can be directed to the corresponding author.

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
