# Peer review of "Evaluation of Direct Antimicrobial Susceptibility Testing of Gram-Negative Bacilli and Staphylococcus aureus from Positive Pediatric Blood Culture Bottles Using BD Phoenix M50"

_microorganisms, 2024, doi:10.3390/microorganisms12081704_

Round 1
Reviewer 1 Report
Comments and Suggestions for Authors
-
Thank you for the opportunity of reviewing the manuscript "Evaluation of Direct Antimicrobial Susceptibility Testing of Gram-Negative Bacilli and Staphylococcus aureus From Positive Pediatric Blood Culture Bottles Using BD Phoenix M50."
The rapidity of provinding a treatment suggestion to the clinitians is indeed important. The study is interesting and I have some suggestions:
Title: The title is clear and descriptive, maybe too long. Ensure the title accurately reflects the scope of the study and shortened it.
Abstract: Add a sentence on the study's significance and potential impact on clinical practice.
Introduction:
- Provide a brief review of relevant literature to justify the study's novelty and importance.
- Clearly state the research question or hypothesis at the end of the introduction.
- Add more references. Here are some suggestions:
- Kumar M, Shergill SPS, Tandel K, Sahai K, Gupta RM. Direct antimicrobial susceptibility testing from positive blood culture bottles in laboratories lacking automated antimicrobial susceptibility testing systems. Med J Armed Forces India. 2019 Oct;75(4):450-457. doi: 10.1016/j.mjafi.2018.08.010.
- ErdoÄŸan G, Karakoç AE, Yücel M, YaÄŸcı S. Pozitif Sinyal Veren Kan Kültürü ÅžiÅŸelerinde EUCAST DoÄŸrudan Hızlı Antimikrobiyal Duyarlılık Testi Yönteminin DeÄŸerlendirilmesi [Evaluation of EUCAST Direct Rapid Antimicrobial Susceptibility Test Method in Blood Culture Bottles with Positive Signal]. Mikrobiyol Bul. 2021 Oct;55(4):626-634. Turkish. doi: 10.5578/mb.20219713.
- Malita, M.A. et al. Cumulative Antibiogram – a Rapid Method to Hinder Transmission of Resistant Bacteria to Oral Cavity of Newborn Babies. Antibiotics 2023, 12, 80. https://doi.org/10.3390/ antibiotics12010080
Materials and Methods
- Include information on the study design (e.g., prospective, retrospective).
- Describe any inclusion and exclusion criteria for the blood culture samples.
- Provide details on the statistical analysis used to compare the methods.
- Mention any ethical approvals obtained for the study, especially since it involves pediatric patients.
Results
- Use more figures and tables to visualize data, such as error rates by organism and antibiotic.
- Ensure all tables and figures are clearly labeled and referenced in the text.
- Highlight any unexpected findings or discrepancies in the results.
Discussion
- Address any limitations of the study, such as sample size or potential biases.
- Suggest directions for future research based on the study's findings.
- Discuss the potential implications for clinical practice, including any benefits and challenges of implementing direct AST in routine diagnostics.
Conclusion
- Ensure the conclusion is concise and directly addresses the research question.
- Reinforce the importance of the study's contributions to the field.
Conflict of Interest and Funding: Include a statement on conflict of interest and disclose any funding sources that supported the research.
Ethical Considerations: Mention any ethical approvals obtained for conducting the study, especially since it involves pediatric patients.
By implementing these detailed suggestions, the manuscript can be significantly improved.
Reviewer 2 Report
Comments and Suggestions for Authors
About the Manuscript ID microorganisms-3162840 title “Evaluation of direct antimicrobial susceptibility testing of gram-negative bacilli and Staphylococcus aureus from positive pediatric blood culture bottles using BD Phoenix M50”
In general, is well written but, will be important to describe better, in more detail, the material and methods. To avoid confusion on the classical method of microorganism identification and the antimicrobial profile analysis in each method.
Just the Enterobacteriales for Enterobacteriaceae and do not use italics
Is a manuscript interesting because is analyzed the quick antimicrobial sensitivity of a positive blood stream infection in children, however, is not complete new, because the same method was made for adults and works in similar way. But must be tested and demonstrated that works with pediatric samples. Therefore. Can be accepted for publication if improve the microbiology methods used for sample processing.
Reviewer 3 Report
Comments and Suggestions for Authors
Line 18. Abstract. Dot missed (end of sentence).
Line 96. Remove double space.
Table 1.
Legend: GNR and GPC. These acronyms are not used in the table, so they should be removed from a legend.
A dot is required after ,,spp,, (e.g. Citrobacter spp.).
I have found no serious linguistic errors.
